# Antibiotic Administration within Two Days after Successful Endoscopic Retrograde Cholangiopancreatography Is Sufficient for Mild and Moderate Acute Cholangitis

**DOI:** 10.3390/jcm11102697

**Published:** 2022-05-10

**Authors:** Sakue Masuda, Kazuya Koizumi, Makomo Makazu, Haruki Uojima, Jun Kubota, Karen Kimura, Takashi Nishino, Chihiro Sumida, Chikamasa Ichita, Akiko Sasaki, Kento Shionoya

**Affiliations:** 1Department of Gastroenterology Medicine Center, Shonan Kamakura General Hospital, 1370-1 Okamoto, Kamakura 247-8533, Japan; kizm2010@gmail.com (K.K.); gonmarota@gmail.com (M.M.); kubojun090@gmail.com (J.K.); lion19930314@gmail.com (K.K.); tkshnshn8@gmail.com (T.N.); ycxcn117@yahoo.co.jp (C.S.); ichikamasa@yahoo.co.jp (C.I.); akikomontblanc@yahoo.co.jp (A.S.); kentosh0812@gmail.com (K.S.); 2Department of Gastroenterology, Internal Medicine, School of Medicine, Kitasato University, Sagamihara 252-0375, Japan; kiruha555@yahoo.co.jp

**Keywords:** antibiotics, antimicrobial stewardship, short-course antimicrobials, cholangitis, endoscopic retrograde cholangiopancreatography

## Abstract

To prevent the increase of resistant bacteria, it is important to minimize the use of antimicrobial agents. Studies have found that administration for ≤3 days after successful endoscopic retrograde cholangiopancreatography (ERCP) is appropriate. Therefore, the present study aimed to verify if administration of antimicrobial agents can be further shortened to ≤2 days after ERCP. We divided 390 patients with mild and moderate cholangitis who underwent technically successful ERCP from January 2018 to June 2020 and had positive blood or bile cultures into two groups: antibiotic therapy within two days of ERCP (short-course therapy, SCT; *n* = 59, 15.1%), and for >3 days (long-course therapy, LCT; *n* = 331, 84.9%). The increased severity after admission and other outcomes were compared between the two groups, and the risk factors for increased severity were verified. There were no between-group differences in patient characteristics. Total length of hospital stay was shorter in SCT than in LCT, and other outcomes in SCT were not significantly different from those in LCT. Being 80 or older was a risk factor for increased severity; however, SCT was not associated with increased severity. Antimicrobial therapy for ≤2 days after successful ERCP is adequate in patients with mild and moderate acute cholangitis.

## 1. Introduction

At present, the incidence of antibiotic-resistant infections is increasing and represents a threat to global health care. One possible reason for this increase in antibiotic resistance is increased antibiotic exposure due to overuse, misuse, or even appropriate use. Prolonged antibiotic treatment can also lead to the development and increase of antibiotic-resistant bacteria [1]. Therefore, it is important to minimize the use of antibiotics in order to reduce the increase in resistant bacteria and the side effects of antibiotics. Furthermore, longer durations of antibiotic treatment are associated with longer hospital stay [2]. This exposes patients at risk to several well-documented complications of prolonged hospitalization, including pneumonia, venous thromboembolism, and muscle loss (especially in elderly patients) [3,4].

Acute cholangitis is a bacterial infection of the bile ducts that can be life-threatening if not diagnosed and treated on time. It is the second most common cause of community-acquired bacteremia and bacteremia in older patients [5,6]. Cholangitis-related mortality rates are relatively high (5–10%) [7,8], and the mortality rate of cholangitis patients who underwent successful endoscopic retrograde cholangiopancreatography (ERCP) is 0–7.2% [9]. The treatment for acute cholangitis mainly includes antimicrobial therapy and biliary decompression according to disease severity, and absence of treatment is associated with a high mortality risk [10]. The most up-to-date and widely used guideline on the subject is the 2018 Tokyo Guidelines (TG18) [9]. TG18 recommends four to seven days of antimicrobial therapy for patients with acute cholangitis once the source of infection has been controlled. However, the evidence level for this recommendation has been graded as low [10]. The national sepsis guideline in the Netherlands is the most progressive on antimicrobial therapy duration in cholangitis, with a recommended therapy duration of ≤3 days after successful biliary drainage [11]. Moreover, recent studies on acute cholangitis suggest that antimicrobial therapy for three days after successful ERCP is sufficient for treatment [11,12]. A large randomized controlled trial on patients with intra-abdominal infections demonstrated that a fixed four-day course of antimicrobial therapy was as effective as a longer, symptom-based treatment duration [13]. For mild or moderate acute cholecystitis, antimicrobial therapy for one day has also been reported to be sufficient after successful cholecystectomy [14,15]. Thus, there is a growing number of reports supporting that short-term administration of antimicrobial therapy is sufficient. In addition, several reports suggest that that even when antimicrobial therapy is ineffective, outcomes for patients with acute cholangitis are not worse if ERCP is successful [16,17].

Therefore, the purpose of this study was to verify whether the duration of antibiotic therapy for patients with mild and moderate acute cholangitis after ERCP can be shortened to ≤2 days.

## 2. Materials and Methods

### 2.1. Study Population

This retrospective observational cohort study was conducted at the Shonan Kamakura General Hospital in Japan. We searched the hospital records of patients treated at the hospital from January 2018 to June 2020 and identified 390 patients with mild and moderate cholangitis who had positive blood or bile cultures and had undergone technically successful ERCP. In principle, blood cultures were collected before antibiotic administration, and bile cultures were collected immediately after the start of the ERCP. We divided the 390 patients into two groups: antibiotic therapy within two days of ERCP (short-course antibiotic therapy, SCT) and antibiotic therapy for ≥3 days (long-course antibiotic therapy, LCT).

Patients who had suffered multiple episodes of cholangitis were included multiple times if the minimum interval between episodes was three months. Patients who had died within 2 days after the initial ERCP, or who were lost to follow-up within 30 days after the initial ERCP, were excluded. Patients who had died within 2 days after the initial ERCP were excluded because they did not have the chance to be treated with antibiotics for more than 2 days (Figure 1).

In our hospital, ampicillin/sulbactam, cefmetazole, ceftriaxone, piperacillin/tazobactam, meropenem, and ciprofloxacin are typically used for the initial treatment. Mild cases were primarily treated with cefmetazole, and other antibiotics were administered based on renal function and previous cultures.

In principle, the dose of each antibacterial agent was as follows: ampicillin/sulbactam (6.0 g/day), piperacillin/tazobactam (13.5 g/day), ceftriaxone (2.0 g/day), cefmetazole (3.0 g/day), meropenem (3.0 g/day), ciprofloxacin (0.6 g/day), levofloxacin (0.5 g/day), and vancomycin (30 mg/body weight kg/day); moreover, the dose of each antibiotic was reduced as needed in patients with reduced GFR.

This was a retrospective study, and there were no criteria for determining the duration of antimicrobial therapy; each physician in charge made the decision.

In cases where plastic stent implantation was needed in ERCP, a single 7-Fr stent was implanted as a rule. When implanting self-expandable metallic stents, stents with diameters of 10 and 8 mm were implanted in the common bile duct and hilar region, respectively.

### 2.2. Ethical Information

All procedures were performed in accordance with the ethical standards established in the 1964 Declaration of Helsinki and its later amendments. The study was reviewed and approved by the institutional review board of the Future Medical Research Center Ethical Committee (IRB no. TGE01849-024, date of approval was 25 November 2021). Due to the retrospective study design, informed consent was obtained from all participants by the opt-out method on our hospital website and in-hospital posting.

### 2.3. Study Outcomes

The primary outcome was an increase in disease severity after hospitalization. Secondary outcomes included national early warning score (NEWS), in-hospital mortality, 30 day mortality, total length of hospital stay, and three-month recurrence—defined as the recurrence of symptoms after complete recovery within 3 months of the disease onset. We reported the results per episode of cholangitis. The three-month recurrence included recurrent cholangitis, cholecystitis, and liver abscess, that could be related to the primary cholangitis episode.

The NEWS was developed by the United Kingdom’s National Early Warning Score Development and Implementation Group in 2012 to assess deteriorating conditions in hospitalized patients, and to predict inpatient death or ICU admission [18]. NEWS measures physiological parameters (systolic blood pressure, pulse rate, respiratory rate, temperature, oxygen saturation), level of consciousness, and oxygen supplementation, all of which are simple and easily accessible [18,19]. NEWS is now widespread in many countries because of its greater ability compared to other early warning scores to identify patients at risk for the composite outcome of cardiac arrest, unexpected ICU admission, and death within 24 h. Most reports place the low-risk group for NEWS at 4 points or less, although one report places it at 3 points or less [20].

For reference, we confirmed the following items: number of days required to break the fever after ERCP, and complications of infective endocarditis within 3 months.

### 2.4. Definitions

The diagnosis and severity of cholangitis were based on the TG18 [21]. An increase in severity was defined as a change from mild to moderate or from moderate to severe, according to the TG18 criteria. Hospital stay was defined as the number of days from the day of admission to the day of discharge or date of death.

Pathogen resistance to the initial antibiotics was defined as a pathogen that was resistant to the initial antibiotics in vitro. Cholangitis is often caused by polymicrobial infections. Blood cultures have low sensitivity and may not detect the causative organism; moreover, bile cultures have low specificity and may detect enteric bacteria that are not the causative organism. This makes it difficult to accurately identify the causative organisms in cholangitis. Therefore, in this study, all the detected bacteria were treated equally as causative organisms.

Clinical success of ERCP was defined as a 50% decrease in the level of total bilirubin or alanine aminotransferase, or normalization within 1 week of ERCP.

Recurrent cholangitis was defined as recurrence of symptoms or laboratory tests after the complete cure of the disease within 3 months after the ERCP. ERCP was performed again in almost all cases of recurrent cholangitis.

The days required to break the fever after ERCP was determined when the temperature remained below 37 °C for 24 h [22].

### 2.5. Statistical Analyses

The Mann-Whitney U-test was used to compare non-normally distributed continuous variables, and the χ^2^ test or Fisher’s exact test was used to compare categorical variables. The Kolmogorov-Smirnov test and the Shapiro–Wilk normality test were used to test the normality of distribution. Multivariate analysis was performed using logistic regression. Variables that were clinically significant or reported in previous studies to be clinically significant were included in the multivariate analysis. Two-tailed *p*-values < 0.05 were considered significant. All statistical analyses were performed using EZR (Saitama Medical Center, Jichi Medical University, Saitama, Japan), which is a graphical user interface for the R statistical software (The R Foundation for Statistical Computing, Vienna, Austria). More precisely, it is a modified version of the R commander, designed to allow additional biostatistical functions [23].

## 3. Results

### 3.1. Patient Characteristics

Table 1 summarizes the patient characteristics. We retrospectively analyzed the data of 390 patients with positive blood or bile cultures who were treated with ERCP. Of these, 59 patients (15.1%) received short-course antibiotic treatment (SCT, ≤2 days) and 331 (84.9%) received long-course antibiotic treatment (LCT, ≥3 days). There were no significant between-group differences in age, sex, cause of cholangitis, disease severity, hyperbilirubinemia, abnormal white blood cells, hypoalbuminemia, NEWS (on admission, just before ERCP), positive blood culture, underlying disease, or patient background. This study included patients with mild and moderate cholangitis, and therefore, did not include patients with severe renal impairment or coagulation abnormalities.

### 3.2. ERCP Findings

Table 2 summarizes the ERCP findings of the study population. The median time from first patient–physician contact to ERCP was longer in the SCT group than in the LCT group (SCT: median, 24 h; range 2–250 h; LCT: median, 10 h; range, 1–120 h; *p* < 0.001). The clinical success rates of ERCP were 94.3 and 95.3% in the SCT and LCT groups, respectively. No significant between-group differences were found with regard to the rate of prior ERCP, clinical success rate of biliary drainage, ERCP drainage procedure, or complications.

### 3.3. Microbiological Data

The laboratory findings of microbial cultures from the patients are summarized in Table 3. The positive rates of blood and bile cultures were 46.4% (13/28) and 100.0% (58/58) in the SCT group, and 49.2% (123/250) and 98.8% (321/325) in the LCT group, respectively. There were no significant differences in the positive rates of blood or bile cultures. A total of 210 patients (53.8%) had polymicrobial infections, for which *Escherichia coli*, *Klebsiella* sp., *Enterococcus* sp., and *Enterobacter* sp. were the most common pathogens.

### 3.4. Antibiotic Therapy

Table 4 summarizes the findings related to antibiotics in the present study. Cefmetazole, ampicillin/sulbactam, and piperacillin/tazobactam were the most commonly used antibiotics. There were no significant differences in the rate of use of each antibiotic. Two patients in the SCT group (3.4%) did not receive any antibiotics.

The median times from the first patient–physician contact to antibiotic administration were 5 and 4 h in the SCT and LCT groups, respectively. The median total duration of antibiotic therapy—including pre-ERCP—was shorter in the SCT than in the LCT group (SCT: median, 2 days; range, 0–12 days; LCT: median, 5 days; range, 3–49; *p* < 0.001).

In the SCT and LCT groups, 23 (39.0%) and 72 (21.8%) patients, respectively, exhibited only pathogens resistant to the initial antibiotics. Of these patients, 38 (52.8%) in the LCT group changed the initial antibiotics to appropriate definitive antibiotic therapy, while those in the SCT group remained on inappropriate antibiotics and completed antimicrobial therapy.

### 3.5. Clinical Outcomes

Table 5 summarizes the clinical outcomes in the present study. Increased severity occurred in 23.7% (14 cases) in the SCT group and in 20.8% (69 cases) in the LCT group. We did not find any significant between-group differences in the increased severity, NEWS (96 h after ERCP, five points or more at five to seven days after ERCP), in-hospital mortality, 30 day mortality, and three-month recurrence.

The median duration of hospitalization was seven days (range, 3–39 days) in the SCT group and eight days (range, 3–120 days) in the LCT group. The duration of hospitalization was significantly shorter in the SCT group compared to the LCT group (*p* = 0.009).

The days required to break the fever after ERCP was significantly longer in the LCT group compared to the SCT group (*p* < 0.001).

No complications of infective endocarditis were identified within three months.

### 3.6. Multivariate Analysis for Increased Severity after Admission

Table 6 summarizes the multivariate analysis for increased severity. Multivariate analysis showed that an age of 80 or more independently predicted the increased severity (odds ratio [OR], 2.16; 95% confidence interval [CI], 1.16–4.05: *p* = 0.016), whereas SCT was not a risk factor. No significant differences were noted in other parameters, including residence in a nursing home, malignant biliary stricture, ERCP within 24 h of first physician contact, diabetes, pathogens resistant to the initial antibiotics, and positive blood culture.

### 3.7. Clinical Outcomes in Positive Blood Culture Group

Table 7 summarizes the clinical outcomes in positive blood culture group. We performed a subgroup analysis focused on blood culture-positive patients and did not find any significant between-group differences in the increased severity, NEWS (96 h after ERCP, 5 points or more at 5 to 7 days after ERCP), in-hospital mortality, 30 day mortality, and three-month recurrence.

## 4. Discussion

In this study, there were no significant differences in the outcomes between the two groups with regard to increased severity, in-hospital mortality, thirty-day mortality, and three-month recurrence. The duration of hospitalization was shorter in the SCT group (≤2 days) than in the LCT group (≥3 days). This result on the duration of hospitalization may be because the duration of the antibiotics treatment itself led to an increased duration of hospitalization. However, the SCT was not associated with worsening of the cholangitis or prolongation of hospital stay. These findings suggest that SCT was not inferior to LCT in patients who underwent successful biliary drainage. Furthermore, the duration of hospitalization was lower in the SCT group. This observation, although statistically significant, may not be clinically meaningful; however, it may suggest that SCT may be useful for preventing complications associated with prolonged hospitalization, such as pneumonia, venous thromboembolism, and muscle loss in the elderly [3,4]. Multivariate analysis showed that an age of 80 or more increased the risk of increased severity, whereas SCT did not make outcomes worse. Several recent studies have acknowledged that SCT does not worsen outcomes—such as mortality and recurrence rates—in patients with successful ERCP. Similar results were reported by Satake et al. in a retrospective study on mild and moderate acute cholangitis and by Haal et al. in a retrospective study on acute cholangitis due to common bile duct stones [11,12]. Van Lent et al. concluded that short-duration (three days) antibiotic treatment for acute cholangitis following adequate biliary duct drainage appeared to be sufficient for treatment [24]. A recent systematic review has reported that short-course antibiotic therapy seems adequate for the treatment of acute cholangitis following successful biliary drainage. However, the review also described that the quality of evidence remains very low due to the low number of patients included, the differences in study design, and the heterogeneity of the definitions used for long- and short-course treatment [9]. The results of these studies suggest that once the source of infection is controlled with biliary drainage, bacteremia is likely to resolve, and the patient may not need further antibiotic therapy [25]. In the present study, we used data from 390 cases, and we conducted a detailed study of short-term prognosis, including not only increased severity of disease, but also NEWS (a new, simple, and easily accessible score for assessing deteriorating conditions and predicting increasing severity of disease). We also conducted long-term prognosis, including the rate of recurrence after three months and the complications of infective endocarditis. Using these data, we showed that SCT is non-inferior to LCT in acute cholangitis, even if the patient had a positive blood culture.

In the present study, more days were required to break the fever in the LCT group than in the SCT group; however, NEWS, the overall score of physical findings including fever, did not differ between the two groups. Although this may seem to indicate that severe cases were more common in LCT, it is possible that clinicians may be overly influenced by only a single item, i.e., fever, to prescribe antibiotics for a longer duration. Van Lent et al. stated that the fear of complications of acute cholangitis drives clinicians to prescribe antibiotics for longer periods of time [24]. Traditionally, physicians have administered antimicrobial therapy to patients with infections until the infection is cured based on clinical and laboratory evidence. This was because they believed that persistent sepsis indicated continued replication of the pathogen. However, recent experimental data suggest that prolonged SIRS may be a reflection of host immune activity rather than an indication of the presence of viable microorganisms. Efforts have therefore begun to shorten the duration of antimicrobial therapy. These efforts have already been successful in other severe infections, such as ventilator-associated pneumonia [13]. Thus, several recent studies have shown that antibiotics can be terminated in the short term, without clinical and laboratory evidence.

Although the level of evidence is low, the TG18 recommends two weeks of antibiotic therapy for cases with Gram-positive cocci (GPC)-positive blood cultures because of concerns about infected endocarditis (IE). However, Gomi et al. validated 6433 patients with cholangitis and found 243 cases with GPC-positive blood cultures, but no complications of IE were observed in those cases. In that study, the overall incidence of IE was 0.26% (17 cases) [26]. In the present study, there were no complications of IE within three months among 40 patients with GPC-positive blood cultures. Although two weeks of antibiotic therapy may be appropriate in patients at risk for IE—such as patients with valvular disease and chronic poor oral hygiene—complications of IE are rare in patients with cholangitis. Therefore, SCT may be appropriate for cholangitis patients, even if their blood cultures are positive for GPC. Doi et al. showed that for acute cholangitis with bacteremia and successful biliary drainage, a shortened total duration (seven days) of antibiotic treatment may be a reasonable option [27]. However, it is still unknown whether the risk of complications from long-term antibiotic therapy should be prioritized over the risk of IE, or vice versa.

The aforementioned results should be interpreted with caution, taking into account some uncertainties. First, this was a single-center retrospective study, and the duration of antimicrobial therapy was determined by each physician in charge. Therefore, patients who were possibly considered to have a severe condition may have been categorized into the LCT group. This means that although there was no significant difference in the severity of the condition according to the TG18 and NEWS between the SCT and LCT groups, potentially severe cases may have been included in the LCT group. Second, because this was a retrospective study, we consider that there are confounding factors that are not included in this study. In addition, because of the small number of patients in the SCT group, rare complications such as IE and liver abscesses have not been adequately evaluated. In particular, data on complications of IE were not examined in detail and should be considered only as a reference. Furthermore, malignant biliary stricture accounted for 102 cases (26.2% of all cases) in this study, and further studies are needed to determine whether SCT is effective for malignant biliary stricture. Third, the rate of pathogen resistance to initial antibiotics was 21.8% in the LCT group. The inefficacy of antibiotics can be approximated as a short duration of antibiotic treatment; therefore, the proportion of patients with pathogen resistance to initial antibiotics (21.8%) in the LCT group may have influenced the results of the present study. However, pathogen resistance to initial antibiotics was also not associated with increased severity in the multivariate analysis. In view of these limitations, we believe that a new randomized controlled study on the duration of antibiotic treatment in acute cholangitis can provide high-quality evidence on the topic. The results of the present study provide a foundation for a future randomized controlled trial, which we plan to conduct to satisfy the need for high-quality evidence.

## 5. Conclusions

This retrospective study indicates that antimicrobial therapy for ≤2 days is sufficient after successful ERCP in patients with mild and moderate acute cholangitis. Prospective studies are needed to confirm our results and to ensure evidence-based recommendations of antibiotic therapy duration in patients with cholangitis. This could ultimately help reduce the unnecessary administration of antibiotics, duration of hospital stays, and associated adverse events.

## Figures and Tables

**Figure 1 jcm-11-02697-f001:**
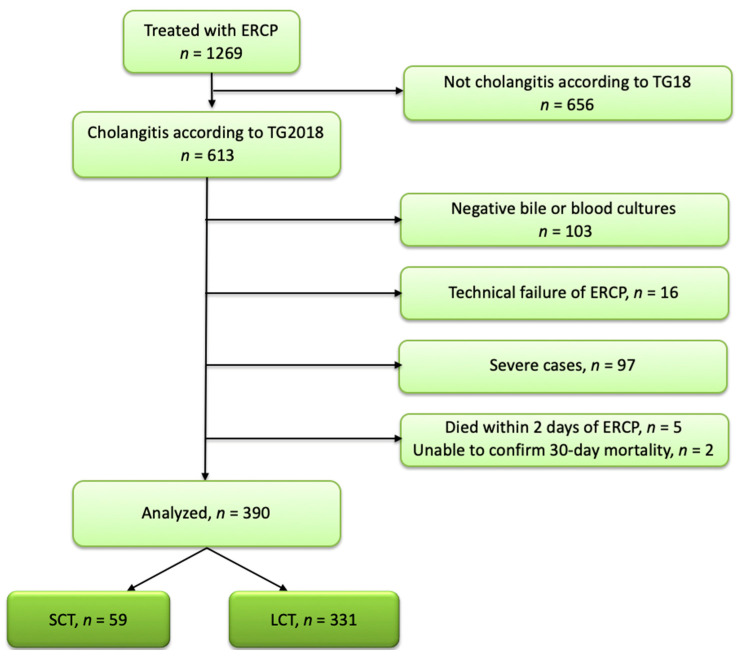
Study population. SCT: short course antibiotic therapy; LCT: long course antibiotic therapy; ERCP: endoscopic retrograde cholangiopancreatography; TG18: Tokyo Guidelines 2018.

**Table 1 jcm-11-02697-t001:** Patient characteristics.

	SCT, *n* = 59 (15.1%)	LCT, *n* = 331 (84.9%)	*p*-Value
Age (median) (range)	81 (26–100)	81 (25–102)	0.837
Sex (male:female)	28:31	172:159	0.579
Cause of cholangitis			
Malignant stricture	17 (28.8%)	85 (25.7%)	0.631
Bile duct stone	36 (61.0%)	216 (65.3%)	0.556
Benign bile duct stricture	3 (5.1%)	8 (2.4%)	0.223
Chronic pancreatitis	1 (1.7%)	4 (1.2%)	0.562
Mirizzi syndrome	0 (0.0%)	9 (2.7%)	0.366
Autoimmune pancreatitis	1 (1.7%)	1 (0.3%)	0.28
Others	1 (1.7%)	8 (2.4%)	>0.99
Severity			
Mild	32 (54.2%)	151 (45.6%)	0.258
Moderate	27 (45.8%)	180 (54.4%)	0.258
Bil ≥ 5.0 (mg/dL)	11 (18.6%)	72 (21.8%)	0.73
WBC < 4000, 12,000 < WBC (/μL)	14 (23.7%)	111 (33.5%)	0.173
Alb < 3.0 (g/dL)	18 (30.5%)	100 (30.2%)	>0.99
NEWS (median) (range)			
On admission	1 (0–7)	1 (0–13)	0.096
Just before ERCP	1 (0–7)	1 (0–13)	0.159
48 h after ERCP	0 (0–5)	0 (0–6)	0.429
Underlying disease			
CKD	5 (8.5%)	26 (7.9%)	0.797
CHF	2 (3.4%)	34 (10.3%)	0.139
LC	3 (5.1%)	15 (4.5%)	0.743
DM	8 (13.6%)	62 (18.7%)	0.461
Malignant tumor	18 (30.5%)	92 (27.8%)	0.642
Patient background			
Nursing home	7 (11.9%)	70 (21.1%)	0.112
Hemodialysis	0 (0.0%)	8 (2.4%)	0.613
Gastrostomy	1 (1.7%)	1 (0.3%)	0.28
Constant placement of urinary catheter	0 (0%)	2 (0.6%)	>0.99
Aspiration pneumonia	0 (0%)	7 (2.1%)	0.601
Immunosuppressant	0 (0%)	6 (1.8%)	0.597
Re-intervention to bile duct stent	11 (18.6%)	65 (19.6%)	>0.99

Some cases overlapped. SCT: short-course antibiotic therapy; LCT: long-course antibiotic therapy; Bil: bilirubin; WBC; white blood cells; Alb: albumin; NEWS: national early warning score; ERCP: endoscopic retrograde cholangiopancreatography; CKD: chronic kidney disease; CHF: chronic heart failure; LC: liver cirrhosis; DM: diabetes mellitus.

**Table 2 jcm-11-02697-t002:** ERCP findings.

	SCT, *n* = 59 (15.1%)	LCT, *n* = 331 (84.9%)	*p*-Value
Median time from first physician contact to ERCP * (hours) (range)	24 (2–250)	10 (1–120)	<0.001
Prior ERCP	22 (37.3%)	137 (41.4%)	0.666
Clinical success of ERCP*1	51/54 (94.3%)	302/317 (95.3%)	0.734
ERCP drainage procedure			
Stent replacement			
Self-expandable metallic stent	10 (16.9%)	37 (11.2%)	0.2
Plastic stent	14 (23.7%)	109 (32.9%)	0.174
ENBD	1 (1.7%)	20 (6.0%)	0.224
Lithotripsy	35 (59.3%)	177 (53.3%)	0.479
Others	1 (1.7%)	2 (0.6%)	0.389
Complications			
Pancreatitis	0 (0.0%)	10 (3.0%)	0.371
Bleeding	2 (3.4%)	8 (2.4%)	0.652
Perforation	0 (0.0%)	2 (0.6%)	>0.99
Cholecystitis	1 (1.7%)	9 (2.7%)	>0.99
Stent migration/early stent obstruction	0 (0.0%)	3 (0.9%)	>0.99
Others	0 (0.0%)	2 (0.6%)	>0.99
Total †	3 (5.1%)	31 (9.4%)	0.45

SCT: short-course antibiotic therapy; LCT: long-course antibiotic therapy; ERCP: endoscopic retrograde cholangiopancreatography; ENBD: endoscopic nasobiliary drainage. * Cases in which the efficacy of ERCP could not be determined were excluded † Some cases overlapped.

**Table 3 jcm-11-02697-t003:** The laboratory findings of microbial cultures.

	SCT, *n* = 59 (15.1%)	LCT, *n* = 331 (84.9%)	*p*-Value
Blood culture			
Positive rate	13/28 (46.4%)	123/250 (49.2%)	0.153
*Escherichia coli*	7 (11.9%)	59 (17.8%)	0.346
*Klebsiella* sp.	2 (3.4%)	33 (10.0%)	0.137
*Enterococcus* sp.	0 (0.0%)	12 (3.6%)	0.227
*Enterobacter* sp.	1 (1.7%)	8 (2.4%)	>0.99
*Citrobacter* sp.	1 (1.7%)	3 (0.9%)	0.483
*Staphylococcus* sp.	0 (0.0%)	7 (2.1%)	0.601
*Streptococcus* sp.	1 (1.7%)	6 (1.8%)	>0.99
*Pseudomonas* sp.	0 (0.0%)	3 (0.9%)	>0.99
*Aeromonas* sp.	1 (1.7%)	13 (3.9%)	0.704
Others	1 (1.7%)	7 (2.1%)	>0.99
Negative	15 (25.4%)	128 (38.7%)	0.057
No culture	31 (52.5%)	81 (24.5%)	<0.001
Bile culture			
Positive rate	58/58 (100.0%)	321/325 (98.8%)	>0.99
*Escherichia coli*	18 (30.5%)	124 (37.5%)	0.378
*Klebsiella* sp.	16 (27.1%)	117 (35.3%)	0.237
*Enterococcus* sp.	29 (49.2%)	123 (37.2%)	0.085
*Enterobacter* sp.	9 (15.3%)	55 (16.6%)	>0.99
*Citrobacter* sp.	8 (13.6%)	22 (6.6%)	0.105
*Staphylococcus* sp.	0 (0.0%)	9 (2.7%)	0.366
*Streptococcus* sp.	7 (11.9%)	40 (12.1%)	>0.99
*Pseudomonas* sp.	2 (3.4%)	22 (6.6%)	0.555
*Aeromonas* sp.	6 (10.2%)	30 (9.1%)	0.807
Others	7 (11.9%)	24 (7.3%)	0.291
Negative	0 (0.0%)	4 (1.2%)	>0.99
No culture	1 (1.7%)	6 (1.8%)	>0.99

Some cases overlapped. SCT: short-course antibiotic therapy; LCT: long-course antibiotic therapy.

**Table 4 jcm-11-02697-t004:** Antibiotic therapy.

	SCT, *n* = 59 (15.1%)	LCT, *n* = 331 (84.9%)	*p*-Value
Initial antimicrobial therapy			
Ampicillin/sulbactam	8 (13.6%)	60 (18.1%)	0.461
Piperacillin/tazobactam	8 (13.6%)	47 (14.2%)	>0.99
Ceftriaxone	3 (5.1%)	17 (5.1%)	>0.99
Cefmetazole	38 (64.4%)	189 (57.1%)	0.319
Meropenem	0 (0.0%)	8 (2.4%)	0.613
Ciprofloxacin	0 (0.0%)	9 (2.7%)	0.366
Levofloxacin	0 (0.0%)	1 (0.3%)	>0.99
Vancomycin	0 (0.0%)	1 (0.3%)	>0.99
Others	0 (0.0%)	1 (0.3%)	>0.99
No antibiotics	2 (3.4%)	0 (0.0%)	0.023
Median time from first physician contact to antibiotic administration (range)	5.0 h	4.0 h	0.039
	(0–96)	(0–53)	
Median total duration of antimicrobial therapy (range)	2 days	5 days	<0.001
	(0–12)	(3–49)	
Only pathogens resistant to the initial antibiotics	23 (39.0%)	72 (21.8%)	0.008

SCT: short-course antibiotic therapy; LCT: long-course antibiotic therapy.

**Table 5 jcm-11-02697-t005:** Clinical outcomes.

	SCT, *n* = 59 (15.1%)	LCT, *n* = 331 (84.9%)	*p*-Value
Duration of hospitalization (days)	7 days (3–34)	7 days (3–120)	0.009
Increased severity	14 (23.7%)	69 (20.8%)	0.607
NEWS 96 h after ERCP 5 points or more at 5 to 7 days after ERCP	0 (0–3) 1/29 (3.4%)	1 (0–15) 11/226 (4.9%)	0.45 >0.99
In-hospital mortality due to cholangitis	0 (0.0%)	5 (1.5%)	>0.99
Thirty-day mortality	1 (1.7%)	11 (3.3%)	>0.99
Three-month recurrence	4/57 (7.0%)	36/313 (11.5%)	0.485
Days required to break the fever after ERCP	0 (0–8)	1 (0–23)	<0.001

SCT: short-course antibiotic therapy; LCT: long-course antibiotic therapy; NEWS: national early warning score; ERCP: endoscopic retrograde cholangiopancreatography.

**Table 6 jcm-11-02697-t006:** Multivariate analysis for increased severity after admission.

	Increased Severity, *n* = 83	No Change in Severity, *n* = 307	Univariate Analysis, *p*-Value	Multivariate Analysis, *p*-Value	Odds Ratio	95%CI
Aged 80 years or more	56 (67.5%)	158 (51.6%)	0.0126	0.016	2.16	1.16–4.05
Nursing home	15 (18.1%)	62 (20.2%)	0.757	0.166		
Malignant biliary stricture	22 (26.5%)	80 (26.1%)	>0.99	0.998		
ERCP within 24 h of first physician contact	65 (78.3%)	231 (75.2%)	0.665	0.116		
Diabetes	14 (16.9%)	56 (18.2%)	0.872	0.779		
Antimicrobials within 2 days of ERCP (SCT)	14 (16.9%)	45 (14.7%)	0.607	0.371		
Only resistant pathogens to the initial antibiotics	15 (18.1%)	80 (26.1%)	0.151	0.345		
Positive blood culture	35/66 (53.0%)	101/211 (47.9%)	0.484	0.742		

Area under the receiver operating characteristic curve: 0.636. ERCP: endoscopic retrograde cholangiopancreatography. SCT: short-course antibiotic therapy.

**Table 7 jcm-11-02697-t007:** Clinical outcomes in positive blood culture group.

	SCT, *n* = 13 (9.6%)	LCT, *n* = 123 (90.4%)	*p*-Value
Duration of hospitalization (days)	6 days (4–13)	8 days (3–120)	0.03
Increased severity	4 (30.8%)	31 (25.2%)	0.74
NEWS			
96 h after ERCP	1 (0–2)	0 (0–15)	0.913
5 or more at 4 to 7 days after ERCP	0/5 (0.0%)	4/88 (4.5%)	>0.99
In-hospital mortality due to cholangitis	0 (0.0%)	3 (2.4%)	>0.99
Thirty-day mortality	0 (0.0%)	4 (3.3%)	>0.99
Three-month recurrence	0 (0.0%)	11 (9.6%)	0.602

SCT: short-course antibiotic therapy; LCT: long-course antibiotic therapy; NEWS: national early warning score; ERCP: endoscopic retrograde cholangiopancreatography.

## Data Availability

The data that support the findings of this study are available from the corresponding author, SM, upon reasonable request. The technical appendix, statistical code, and dataset are available from the corresponding author upon request. No additional data are available.

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
