# Peer review of "Antibiotic Administration within Two Days after Successful Endoscopic Retrograde Cholangiopancreatography Is Sufficient for Mild and Moderate Acute Cholangitis"

_jcm, 2022, doi:10.3390/jcm11102697_

Round 1

Reviewer 1 Report

The most up-to-date and widely used guideline is 4–7 days of antimicrobial therapy for patients with acute cholangitis once the source of infection has been controlled. The national sepsis guideline in the Netherlands is most progressive on antimicrobial therapy duration in cholangitis, with a recommended therapy duration of ≤3 days after successful biliary drainage. The authors investigated whether the duration of antibiotic therapy for patients with mild and moderate acute cholangitis after ERCP can be shortened to ≤2 days. They concluded that antimicrobial therapy for ≤2 days after successful ERCP is adequate in patients with mild and moderate acute cholangitis.

Overall, interesting, important and well-designed study. There are several, mostly minor, issues that need to be corrected before possible publication. I have several suggestions and objections to improve the study:

  1. The authors listed antibiotics that were used in their institution. Please provide exact doses for each antibiotic listed.
  2. The authors stated that the institutional review board of Future Medical Research Center Ethical Committee (IRB no. TGE01849-024) approved the study. Please add a date of approval next to the reference number.
  3. Study design - As it is visible from Figure 1 the authors divided the patients into two groups:  SCT- short course antibiotic therapy and LCT- long course antibiotic therapy. This should be described in methodology. Also, they should describe what was considered as short and what was definition of a long antibiotic therapy, how many days. Also, it is unclear how the authors decided on the duration of antibiotic treatment?
  4. Figure 1 – There is no need to repeat the same text from methodology in description of the Figure. Also, there is an error in this Figure - the ‘TG18’ is underlined in red. Please revise.
  5. Statistical analysis – Which statistical test was used to test normality of distribution? This should be mentioned in the statistical analysis paragraph.
  6. Tables 1, 5 and 7 – An abbreviation ERCP should be mentioned in legend of the Table.
  7. Table 6. – An abbreviation SCT should be mentioned in the legend of the Table.
  8. Single center design should be mentioned under the limitations of the study.

Reviewer 2 Report

I would be really interested to read the results of the prospective study they plan to start after this retrospective analysis, duration of antibiotic therapy is surely something clinitians argue about every day with infective disease specialists, demonstrating safety even if we are reducing time of administration would be paramount

Reviewer 3 Report

High-quality Clinical data on antibiotic duration in acute cholangitis are still very scarce. The authors successfully conducted a retrospective study on antibiotic treatment in such patients. The study reflects a high standard of care in the study center. However, a fewmethodical flaws might require a  workover.

Major comments:

  • The choice of the primary endpoint is problematic, since antibiotic duration (most of which are mandatorily administered i.v.) itself will lead to increased duration of hospital admission
  • Recurrent cholangitis was not defined. This must be adressed. It will not suffice to make this a clinical diagnosis, but therapy of recurrent cholangitis must be adressed as well. Did these patients undergo ERCP anew?
  • Short CT population was significantly smaller than Long CT population. Short CT cases therefore appear to be less complicated. How is this reflected by the data? Important laboratory data are missing (Bilirubin, CRP, leukocytes, Kidney funciton, coagulation)
  • In line with this, the single parameters constituting NEWS criteria should be presented among both groups as well.
  • Since Resolution of fever was significantly longer in the long CT group, the study mainly reports a selection phenomenon. Therefore, the conclusion does not withstand and must be specified. Short CT can only be safely applied in patients with immediate resolution of fever, lest further evidence is presented.

Minor comments:

  • There were only few cases of malignant cholangitis, while main pathology were gall stones. It must be discussed that these patients likely require shorter CT than patients with malignant stenosis.
  • Median duration os hospitalization was equal among both groups. Therefore, mean values should be presented; otherwise, statistical difference cannot be postulated. To this end, e.g. boxplots are recommended.
  • page 7, line 192-194: In the SCT and LCT groups, 23 (39.0%) and 72 (21.8%) patients, respectively, exhibited only pathogens resistant to the initial antibiotics. Of these patients, 38 (52.8%) changed the initial antibiotics to appropriate definitive antibiotic therapy. Mathematically, 38 (52.8%) refers exclusively to 72 (21.8%) patients. Please explain. All numbers and percentages throughout the manuscript should be double-checked.
